# Deep Neural Networks for Accurate Depth Estimation with Latent Space Features

**DOI:** 10.3390/biomimetics9120747

**Published:** 2024-12-09

**Authors:** Siddiqui Muhammad Yasir, Hyunsik Ahn

**Affiliations:** 1Department of Mechanical System Engineering, Tongmyong University, Busan 48520, Republic of Korea; siddiqui@seoultech.ac.kr; 2School of Artificial Intelligence, Tongmyong University, Busan 48520, Republic of Korea

**Keywords:** human–robot interaction, deep learning in robotics, computer vision for automation, depth estimation, multi-scope vision

## Abstract

Depth estimation plays a pivotal role in advancing human–robot interactions, especially in indoor environments where accurate 3D scene reconstruction is essential for tasks like navigation and object handling. Monocular depth estimation, which relies on a single RGB camera, offers a more affordable solution compared to traditional methods that use stereo cameras or LiDAR. However, despite recent progress, many monocular approaches struggle with accurately defining depth boundaries, leading to less precise reconstructions. In response to these challenges, this study introduces a novel depth estimation framework that leverages latent space features within a deep convolutional neural network to enhance the precision of monocular depth maps. The proposed model features dual encoder–decoder architecture, enabling both color-to-depth and depth-to-depth transformations. This structure allows for refined depth estimation through latent space encoding. To further improve the accuracy of depth boundaries and local features, a new loss function is introduced. This function combines latent loss with gradient loss, helping the model maintain the integrity of depth boundaries. The framework is thoroughly tested using the NYU Depth V2 dataset, where it sets a new benchmark, particularly excelling in complex indoor scenarios. The results clearly show that this approach effectively reduces depth ambiguities and blurring, making it a promising solution for applications in human–robot interaction and 3D scene reconstruction.

## 1. Introduction

In human–robot interactions, comprehending the spatial relationships of 3-dimensional (3D) surroundings is a critical perceptual undertaking for various robotic applications, encompassing manipulation, exploration, and navigation [1]. Accurate depth perception is typically required by robots to evade obstructions and handle objects. In industrial settings, moving autonomous agents often possess a color camera for surveillance purposes. However, the depth estimation frequently necessitates specialized equipment, i.e., stereo cameras, and light or time-of-flight sensors, which are relatively costly compared with single RGB cameras. Although researchers have made progress in monocular depth perception, considerably more remains to be achieved [2]. In recent times, the techniques of estimating depth from a single monocular image are increasingly important in computer vision applications, including 3D scene modeling and reconstruction, and autonomous driving systems, as it provides valuable insights into scene vanishing points and horizontal boundaries [1,2]. The stereo-matching principle is a fundamental technique for estimating depth using multiple cameras. Consequently, stereo matching can generate a disparity map that depicts positional differences linking corresponding pixels in stereo images [3]. When multiple images captured at aligned camera positions are used for depth estimation, it is referred to as multi-scope vision, analogous to stereo vision, which uses two horizontally aligned images [4].

Recently, accurate depth estimation has become a critical capability in robotics, particularly for tasks that involve human–robot interaction (HRI), such as object manipulation, navigation, and environment understanding. These methods utilize a color image as input to convert it into a corresponding depth image, with the aim of addressing the limitations of statistical feature-based methods in accommodating diverse geometric structures, particularly in complex indoor environments. This approach aims to improve depth estimation in complex environments [5]. Despite its advantages, monocular depth estimation faces significant challenges, including depth ambiguity and difficulty in preserving depth boundary details. These limitations have driven research toward designing more robust and precise models capable of extracting meaningful depth information from single images [6,7]. Traditional methods for depth estimation, such as stereo matching or structure-from-motion, rely on multiple input images or specialized sensor setups to derive depth information. While effective, these methods are resource-intensive and unsuitable for lightweight systems in indoor environments [8]. Statistical feature-based approaches, leveraging edge detection and segmentation, sought to address these limitations but were constrained by their reliance on hand-crafted features, which often fail in complex indoor scenarios [9]. Deep learning has since emerged as a transformative approach, with convolutional neural networks (CNNs) demonstrating remarkable success in capturing geometric structures and enhancing depth prediction accuracy. Recent advances in deep neural networks for monocular depth estimation have focused on leveraging feature representations at multiple scales to improve both local and global depth predictions [10]. However, the existing approaches often struggle with preserving depth boundary accuracy, leading to blurred edges and diminished performance in challenging scenes with occlusions or variable lighting [11]. Furthermore, many current models exhibit limited generalizability, as their performance is strongly tied to the distribution of the training data. This gap underscores the need for architecture that can effectively encode depth-specific features while generalizing across diverse environments. The proposed research builds on these foundations by introducing a dual encoder–decoder architecture designed to address the identified gaps. Unlike conventional approaches, our method incorporates a novel latent loss function that emphasizes structural consistency in feature space, alongside a gradient loss function to enhance boundary sharpness. By combining color-to-depth and depth-to-depth transformations, the model achieves improved precision in depth estimation while maintaining computational efficiency. These innovations, as elaborated in the subsequent sections, aim to overcome the limitations of previous methods and provide a robust solution for indoor HRI applications.

The paper proposes a lightweight human–robot interaction system for depth estimation from a single RGB image. It uses features extracted from the latent space network to guide the learning process of the RGB image-to-depth relationship. These encoded features contain a geometrical structure that is compactly relevant to the scene’s depth layout, effectively sharpening the depth boundaries. The proposed method minimizes depth ambiguity in homogeneous regions while blurring artifacts at depth boundaries. This biomimetic approach emphasizes low-level visual perception through skip connection, comparable to the primary visual cortex’s basic processing. The main contributions of the proposed method are summarized as follows:-We propose a human–robot interaction system for a monocular depth estimation autoencoder network to effectively learn the complex process of transforming a color image into a depth image. This is unlike previous approaches that relied on the concept of perceptual loss.-The proposed technique aims to discover the process of “generation” from latent space rather than using a “classification” strategy to refine the estimated depth information.-In contrast to other techniques, the proposed method is reasonably accurate because the network design only utilizes skip connections in residual blocks rather than feature branches, leading to effective results.

This paper’s summary is organized as follows: Section 2, a comparison of comparable studies is reviewed. Section 3 provides a thorough explanation of the proposed human–robot interaction system for monocular depth estimation utilizing human–robot interaction to perceive an indoor environment. In Section 4, experimental findings are illustrated using a benchmark dataset. In Section 5, the results and conclusion are presented.

## 2. Related Work

This section explains the methods that integrate data to build depth maps for service robot systems after examining prior technologies for obtaining images for depth estimates.

Depth estimation has emerged as a critical area of research, driven by the demand for precise and efficient methods across diverse applications. Early efforts by Fang et al. [12] established a robust evaluation criterion for active vision systems, offering a foundational framework for depth estimation. López-Nicolás et al. [13] extended this line of inquiry to robotic navigation, proposing an innovative visual servoing approach for mobile robots with fixed monocular vision systems. Similarly, Sabnis et al. [14] explored the optical properties of cameras, leveraging defocus blur to achieve enhanced depth accuracy. This work highlighted the interplay of hardware features and computational techniques for depth sensing. In the medical field, Turan et al. [15] demonstrated the applicability of monocular depth estimation to endoscopic capsule robots, enabling real-time odometry and depth measurement without external supervision. Jin et al. [16] applied depth estimation to humanoid robotics, introducing a progressive approximation (PA)-based cyclic learning framework that adapts to specific behavioral tasks. Xiao et al. [17], drawing inspiration from human tactile sensing, employed a deep recurrent neural network (DRNN) along with long short-term memory (LSTM) to improve tumor depth estimation in soft tissues, underscoring the versatility of deep learning in specialized applications.

Advancements in human–robot interaction (HRI) have further driven the field. Cheng et al. [18] proposed a modular framework that integrates RGB images and human pose estimation to overcome the limitations of depth sensors in dynamic environments. Yu et al. [19,20] focused on disparity estimation for electric inspection robotics by developing a lightweight neural network combining PSMNet and cutting-edge optimization techniques. Wang et al. [21] emphasized the extraction of scene structures and motion characteristics from monocular image sequences, demonstrating the potential of monocular systems in complex scenarios. Shimada et al. [22] addressed the challenge of 3D hand pose estimation and shape refinement using monocular sequences, bypassing the need for explicit depth data. Other innovative methodologies have also surfaced. Pan et al. [23] presented an efficient algorithm for estimating the relative pose of cooperative space targets, integrating multi-target tracking with the Levenberg–Marquardt method (LMM) for rapid convergence. Gysel et al. [24] proposed a latent vector space model capable of handling outliers and learning latent representations by marginalizing changes in depth maps. Kashyap et al. [25] developed an approach to estimate camera motion parameters directly from optic flow, paving the way for improved motion analysis. Reading et al. [26] introduced CaDDN, a fully differentiable method for the simultaneous object detection and depth estimation, demonstrating the synergy of these two tasks in unified frameworks.

Recent efforts have delved deeper into the integration of deep neural networks and latent space features. For example, Pei [27] proposed the Multi-Scale Features Network (MSFNet), incorporating Enhanced Diverse Attention (EDA) and Up-Sample-Stage Fusion (USF) modules for superior depth estimation. However, challenges persist in unsupervised frameworks, particularly in adverse conditions like nighttime or rainy scenes, where traditional photometric consistency assumptions fail. Zhao et al. [28] tackled this issue using an image transfer-based domain adaptation strategy tailored for human–robot interaction in such challenging scenarios. Guo et al. [29] emphasized reducing computational complexity while maintaining high accuracy, a crucial goal for resource-constrained applications.

Collectively, the critical role of deep neural networks in depth estimation across domains ranges from robotics to medical imaging. However, despite these advancements, challenges such as handling extreme environmental variations, computational constraints, clearly revealing depth boundaries, blurry restoration, and real-time adaptability remain. This paper presents a straightforward method for depth estimation using a single RGB image that addresses the issue of blurring artifacts in depth edges. The research builds on these foundations by exploring latent space feature extraction within deep neural networks, aiming to further enhance the accuracy and robustness of depth estimation in diverse applications. The technical specifications and architecture are covered in the next section.

## 3. Proposed Monocular Depth Estimation

This section discusses the proposed monocular depth estimation model and its training methodology. The goal is to improve depth information for a 3D environment model that can understand and serve humans in different ways. This research explores the generative process of depth arrangement from a monochromatic image. A deep convolutional neural network is used to encode RGB to depth relationship in latent space, enhancing the quality of the depth map. The model’s structure is presented, followed by a detailed explanation of the depth estimation procedure using the training approach. The loss function used is explained, including data loss, latent loss, and gradient loss.

### 3.1. Depth Estimation Deep Learning Model

The proposed depth estimation architecture contains depth-to-depth and depth-to-color networks. Both networks have a similar structure with three main elements, an encoder, ResBlocks, and a decoder. The general layout is depicted in Figure 1. The image used as an input is squeezed into latent features by multiple ResBlocks on the encoder side, with a modified version from the original residual network shown in Figure 2.

As is common knowledge in brain science, the primary visual cortex (V1) detects basic visual characteristics such as shape, color, contrast ratio, and line direction, and the secondary visual cortex (V2) uses the detection results of the V1 to recognize a higher level of visual perception such as depth and relationships between objects. Therefore, for detecting the edge of depth more clearly, the role of the V1 perceiving the core visual information is important. The skip connection of the residual block in this approach has the effect of highlighting the core visual information of an image.

By layering a sufficient number of ResBlocks, latent features implicitly encode characteristics for generating depth. The small spatial size of these highly encoded latent features holds necessary information used to reconstruct the target image, specifically the depth map. Batch normalization and ReLU layers follow every convolution layer except the last output layer. On the decoder side, the feature map size is doubled through up-sampling using bilinear interpolation. The depth map is effectively produced by the symmetric decoder using the latent features.

The proposed approach captures internal correlation within spatial dimensions through feature maps, using skip connections to bring back local details [30]. It uses learned latent features related to depth generation to capture specific elements and the overall structure of the depth map. These characteristics influence the outcome, which resembles the real depth map. The guided network implicitly enhances the scheme, allowing it to detect depth boundaries in the estimated results, even in complex outdoor settings.

The comprehensive structure of the proposed network can be found in Figure 2. The sizes of the convolution filters vary from 3 × 3 to 9 × 9 based on the convolution layers they are used in. By utilizing the latent features learned during the depth generation process, the proposed network can reconstruct both local and global details of the depth map output. A thorough description of the training approach for these two networks is explained in Table 1.

### 3.2. Latent Loss Functions

The latent loss function is designed to compare feature representations rather than directly compare pixel-level outputs of the predicted depth map and ground truth. By summing up all spatial locations and feature layers, it evaluates the degree of alignment between *G_j_*(*y*) and *G*(*y**) in the feature space, effectively assessing how well the predicted depth map matches the ground truth in terms of higher-level structural and semantic similarities [31].
(1)$$L_{l}\left(G\left(y\right),G\left(y^{*}\right)\right)\;=\underset{j}{\sum }\;\left(\frac{1}{N_{j}}\;\overset{N_{j}}{\underset{k}{\sum }}\frac{{\left|\left|\;G_{j}\left(y_{k}\right)-G_{j}\left(y_{k}^{*}\right)\right|\right|}^{2}}{2}\;\right)$$

*G_j_*(*y*) and *G*(*y**) refer to feature representations derived from the predicted and ground truth depth maps using guided network *G*. Moreover, *j* indexes the feature layers in the guided network; *k* indexes the spatial locations within a feature layer; *N_j_* represents the total number of spatial locations in layer *j*; and || ||^2^ is squared *L*_2-norm_, which measures the Euclidean distance between feature representations at the corresponding locations.

The objective of this latent loss function is to enforce alignment linking between the predicted and ground truth depth maps at the feature representation level. This approach shifts the focus from traditional pixel-level supervision to a feature–space comparison, enabling the network to be trained in a more robust and semantically meaningful interpretation of depth. By operating in the latent space of the guided network’s topmost encoded layer, this loss facilitates a more direct and efficient supervision mechanism for the decoder, allowing it to generate high-quality depth maps from the latent space. In this research, the latent loss function plays a critical role in achieving perceptual consistency in depth estimation. Unlike conventional classification-based or regression-based losses, which might fail to capture nuanced structural and contextual information, the latent loss provides a feature-level perspective that aligns well with the end task of generating perceptually accurate depth maps. By leveraging *G* to extract dense features, this design ensures that the predicted (restore depth within a depth-to-depth autoencoder framework) depth maps maintain fidelity to the ground truth, even in challenging scenarios such as scenes with complex lighting or occlusions.

### 3.3. Gradiant Loss

The objective of the gradient loss function is to enforce gradient consistency between the predicted and actual depth maps. By comparing the horizontal and vertical gradients separately, the method ensures that the predicted map captures subtle variations in depth, especially along edges and boundaries where depth discontinuities often occur. Incorporating this gradient-based approach serves to address a critical limitation in depth estimation tasks, that is, the difficulty of accurately reconstructing boundary details where neighboring pixels may have starkly different depth values. By aligning the gradients of the predicted and true depth maps, the loss function prioritizes accurate boundary delineation, leading to sharper and more precise depth predictions to be computed as follows:(2)$$L_{gd}\left(y,y^{*}\right)\;=\frac{1}{N}\overset{N}{\underset{i}{\sum }}\;\left|y_{h,i}-y_{h,i}^{*}\right|+\left|y_{v,i}-y_{v,i}^{*}\right|$$

The proposed formula for gradient loss, *L_gd_*_(*y*,*y**)_, portrays a critical role in improving the accuracy of depth estimation, particularly at depth boundaries which are often challenging regions to resolve. This formula incorporates gradient information at both the image and feature levels, ensuring that the predicted depth map aligns more closely with the ground truth by emphasizing local changes and edge details. In addition, *y_h_*_,*i*_ and *y_v_*_,*i*_ denote both (horizontal and vertical) gradient values of the predicted depth map at the *i*-th pixel. Likewise, *y*^*^_*h*,*i*_ and *y*^*^_*v*,*i*_ represent the corresponding (horizontal and vertical) gradient values derived from the ground truth depth map. *N* is the total number of pixels in the depth map.

Furthermore, the gradient loss also extends to the feature level by evaluating the gradients of the encoded features collectively. This additional layer of gradient-based consistency helps refine the model’s representation and understand depth transitions, resulting in better generalization and improved performance. In the context of our research, the inclusion of this loss function contributes to enhancing the model’s ability to resolve fine details in complex scenes. It ensures that the predicted depth maps not only maintain global consistency but also accurately capture local variations, particularly at object edges. This dual-level gradient alignment—at both the image and feature levels—provides a robust mechanism for addressing one of the key challenges in depth estimation and improves the overall quality of the predictions.
(3)$$L_{gl}\left(G\left(y\right),G\left(y^{*}\right)\right)\;=\sum_{j}\;\left(\frac{1}{N_{j}}\;\sum_{k}^{N_{j}}\left|\;G_{h,j}\left(y_{k}\right)-G_{h,j}\left(y_{k}^{*}\right)\right|+\left|G_{v,j}\left(y_{k}\right)-G_{v,j}\left(y_{k}^{*}\right)\right|\right)$$
where *G*_*h*,*j*_ and *G*_*v*,*j*_ represent the gradient-encoded features as specific inputs in both (horizontal and vertical) directions, respectively. The depth map’s components can be effectively enhanced by incorporating both (*G*_*h*,*j*_ and *G*_*v*,*j*_) terms into the ultimate loss function; thus, the depth map’s high-frequency components can be effectively enhanced. A key benefit is that the encoded features within the proposed network effectively represent the depth structure in a condensed manner across multiple scales, leading to significant assistance from their gradients in enhancing the clarity of depth boundaries, as illustrated in Figure 4. Hence, the gradient-encoded features proposed in this article are supposed to be helpful in effectively recovering depth boundaries.

To summarize, the extracted features from the guided network’s latent space enable the learning of the intricate color-depth relationship in the proposed method (refer to (1) and (2)). The depth layout’s core structure can be accurately reconstructed from just one monocular image due to its inherent properties of condensed depth generation. Additionally, the gradients of these encoded features have shown to be effective in recovering the depth boundary, a task that has proven challenging for previous techniques.

## 4. Experiment

We trained our model using the original NYU Depth v2 [27] dataset collected indoors. The unprocessed datasets have many extra pictures gathered from identical locations, similar to those in the popular smaller datasets, but with no prior editing. More precisely, areas that do not have a depth measurement are left blank. However, our model is inherently equipped to deal with such gaps, and its need for a substantial training set makes these original distributions valuable sources of data.

### 4.1. NYU v2 Depth Dataset

The NYU Depth v2 dataset [27] is widely employed in depth estimation analysis and consists of video recordings from 464 indoor scenes. For the purposes of experimentation, the dataset is split into 249 scenes for training and 215 scenes for testing. The RGB input images are resized from their original dimensions of 640 × 480 to 320 × 240 to standardize the input data. Thus, each depth frame is aligned with the nearest RGB frame in temporal proximity. This process necessitates discarding depth frames where multiple depth images correspond to the same RGB frame to maintain one-to-one correspondence. The camera projection parameters provided with the dataset are operated to align RGB and depth frames. Additionally, areas in depth images with invalid values—caused by reflective surfaces or windows—are masked out. The training set contains a total of 120,000 unique images. To address imbalances in the representation of scenes, the images are redistributed, resulting in an expanded training set of 220,000 images, ensuring approximately 1,200 images per scene. For evaluation, the model is tested on a subset of the NYU Depth v2 test set comprising 694 images, where missing depth values have been filled in. An example pair of RGB and depth images is depicted in Figure 3.

This detailed dataset preparation process is critical for ensuring the robustness and fairness of the models trained for depth estimation or related tasks. By aligning the RGB and depth images, excluding invalid pixels, and balancing the training data across scenes, the dataset supports the development of models that generalize well to diverse indoor environments. In this research, the NYU Depth dataset serves as the foundation for training and evaluating models, enabling a systematic investigation of techniques for depth estimation and scene understanding in indoor settings. Simple RGB and ground truth examples are detailed below.

In this research, a two-stage training process was employed to develop and refine the network. Initially, the coarse network was trained using stochastic gradient descent (SGD) on 2 million samples, with a batch size of 32. Following this phase, a fine network was trained on 1.5 million samples, with its parameters fixed and guided by outputs of pre-trained coarse network. For the coarse network, learning rates were carefully calibrated as follows: convolutional layers (layers 1 through 5) were assigned a rate of 0.001, while the fully connected layers (layers 6 and 7) used a significantly higher rate of 0.1. In contrast, the fine network adopted distinct learning rates for its layers as follows: layers 1 and 3 utilized a learning rate of 0.001, while layer 2 employed 0.01. These learning rates were not arbitrarily chosen but were systematically fine-tuned using a validation set through a process of iterative adjustment. After finalizing the learning rate configurations, the validation set was merged with the training data to ensure robust final evaluations. Moreover, the scale of all learning rates was subsequently adjusted by a factor of 5 to optimize performance further. Both networks were trained with a momentum parameter of 0.9, which enhances optimization by accelerating convergence and mitigating oscillations. The coarse network training spanned 38 h, while the fine network training leveraged the computational power of an NVIDIA GTX 1080Ti GPU, significantly expediting the process. This methodical approach underscores the research’s emphasis on optimizing learning parameters to enhance network performance. By systematically fine-tuning the learning rates and employing a staged training framework, the study demonstrates a robust methodology for leveraging pre-trained models and refining subsequent networks for specific tasks.

### 4.2. Performance Evaluation

To demonstrate the generalizability of the proposed approach, we evaluated its performance on the NYU Depth v2 dataset, showcasing several examples of predicted depth estimation in Figure 4. The results illustrate that the proposed method markedly enhances the interpretation of depth in indoor scenes. Furthermore, the training using the NYU Depth v2 dataset reveals strong cross-dataset performance, as it can generate accurate depth maps for the related datasets without requiring additional fine-tuning.

In addition, we examined the computational efficiency of the method by measuring the time required to estimate depth maps from individual input images. These findings are summarized in Table 2. For consistency in evaluation, input images were uniformly resized to dimensions of 512 × 256 pixels. Our method’s performance was compared against existing techniques: a conditional generative adversarial network (GAN) developed by Zhang et al. [32] and a Convolutional Neural Network (CNN) approach introduced by Eigen et al. [33] and Sihaeng et al. [34]. This analysis serves a dual purpose within the scope of the research; initially, it validates the effectiveness and adaptability of the proposed depth estimation model across datasets, and then, it highlights its computational efficiency in relation to state-of-the-art approaches. It presents both qualitative and quantitative comparisons, both recognized for their relatively fast processing speeds. Remarkably, our proposed method operates with exceptional efficiency. Consequently, we posit that our network architecture has substantial potential for application in various human–robot interaction systems.

**Figure 4 biomimetics-09-00747-f004:**
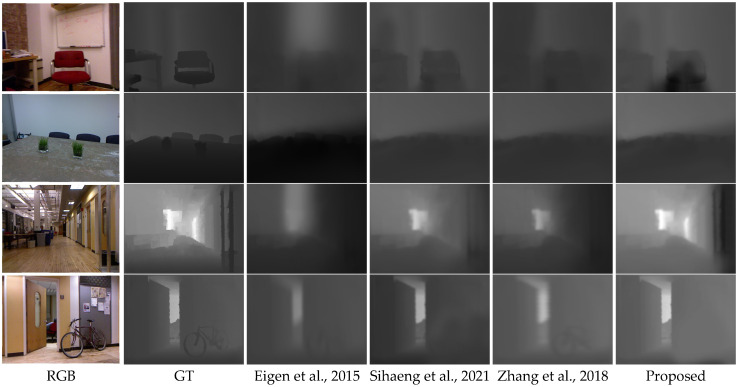
In a visual comparison of the generated depth outcomes, the depth boundaries are anticipated, whereas other methodologies yield hazy and blurry predictions [32,33,34].

In this analysis, our proposed method was evaluated against state-of-the-art techniques using the NYU v2 dataset [27], with the corresponding results summarized in Table 2. The findings demonstrate that our approach achieves superior performance across all evaluation metrics. By leveraging latent features extracted from our guided network and their associated gradients, the proposed method effectively restores depth boundaries, thereby mitigating depth ambiguities and enhancing overall performance. Notably, the proposed method achieves a substantial reduction in root mean squared error (RMSE) as explained in Equation (4), with a 54.13%, 8.37%, and 29.49% improvement compared to the method introduced by Eigen et al. [33], Sihaeng et al. [34] and Zhang et al. [32]. These results underscore the effectiveness of the proposed approach in reconstructing the depth map layout of indoor environments, thereby advancing performance in human–robot interaction systems.
(4)$$RMSE=\sqrt{\frac{1}{\left|N\right|}\;\sum_{y\epsilon T}{\left|\left|y-y^{*}\right|\right|}^{2}}$$

To evaluate the effectiveness and robustness of the proposed method, a comparative analysis was conducted against the standard approaches introduced by Saxena et al. [35]. This study sought to enhance monocular depth estimation by regulating the shift in an RGB camera and capturing images at carefully controlled orientations. The outcomes of three methodologies—Zhang et al. [32], Eigen et al. [33] and Sihaeng et al. [34]—are presented in Figure 4 to provide a qualitative assessment of performance. Ground truth samples are included to enhance visualization and facilitate a more comprehensive evaluation.

In contrast to earlier methods, the proposed approach successfully recovers the depth boundary, particularly revealing the minor sign with a significant difference in Figure 4 that other methods are unable to restore. The object boundaries are also successfully restored with clear corresponding regions. Based on these comparisons, the proposed method can provide reliable depth estimation using a single RGB image. The performance is quantitatively evaluated using the well-known RMSE for depth estimation.

### 4.3. Discussion

The issue of depth perception is a crucial aspect of computer vision, which has been the focus of attention of numerous researchers, resulting in significant advances in recent decades. However, most of the work carried out in this field, such as stereopsis, has relied on using multiple image geometric cues to determine depth. In contrast, single-image cues provide a largely independent source of information, which has not been extensively explored until now. Given the importance of depth and shape perception in various applications, including object recognition, robot grasping, navigation, image compositing, and video retrieval, we believe that monocular depth estimation can significantly enhance these applications, particularly in cases where only a single image of a scene is available. We have developed an algorithm to infer detailed depth estimation from a single still image. The proposed method surpasses previous methods in both quantitative accuracy and visual quality, emphasizing both latent loss and gradient loss. The model assumes an environment consisting of multiple small scene planes and does not explicitly assume the scene’s structure, unlike Delage et al. [36] and Hoiem et al. [37], who assume vertical surfaces on a horizontal floor. This allows the model to generalize distinctly for scenes with significant non-vertical structures.

In a few situations, the proposed model underperformed and produced results that were completely different from the ground reality. The failure possibilities of the anticipated depth estimate are shown in Figure 5. We discovered via our research that dark, low light, and ground truth without details are scenarios that are very difficult to anticipate; thus, an environment with defined boundaries is required to process an accurate prediction. To determine the effectiveness of our approach in real-world human–robot interaction system applications, images captured by the proposed technique are not perfectly calibrated and resolved, resulting in additional noise in correlation of depth estimation. In such cases, single image depth estimation, which is more robust than stereo estimation, is preferred. However, in industrial environments, this approach can be improved for the better reconstruction of a 3D modeling of an indoor environment for human–robot interactions.

## 5. Conclusions

The paper presents a model to estimate the depth map from a single monocular image; the model proposed encoded features from a latent space network to guide the depth estimation process. It emphasizes both latent loss and gradient loss with residual blocks as primary visual perception. The proposed model produces clean boundaries, making it suitable for the 3D modeling of scenes. Experimental findings demonstrate outstanding performance in depth estimation for scene understanding and navigation in indoor environments for human–robot interaction.

## Figures and Tables

**Figure 1 biomimetics-09-00747-f001:**
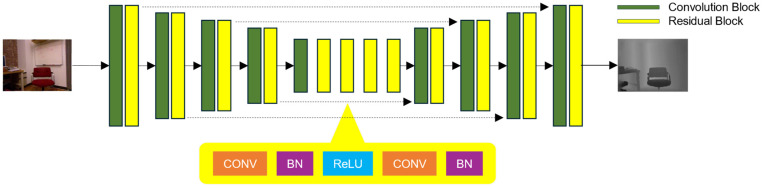
General structure of the proposed depth estimation model. It consists of both convolution and de-convolution layers. In order to understand how color and depth are related in each image effectively, the network is trained with a loss function that includes features from the latent space of the network.

**Figure 2 biomimetics-09-00747-f002:**
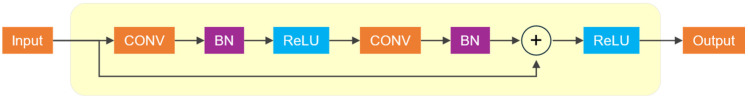
Detailed view of the residual block and its functionality explained in detail.

**Figure 3 biomimetics-09-00747-f003:**
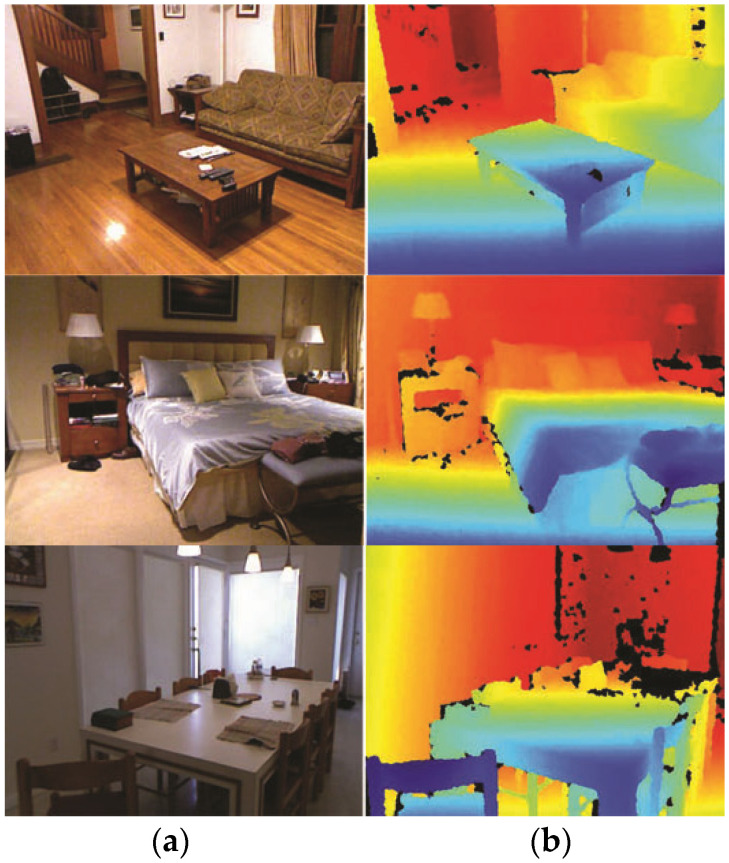
An example of a dataset RGB image and ground truth depth map. (**a**) A single RGB image, and (**b**) the corresponding ground truth.

**Figure 5 biomimetics-09-00747-f005:**
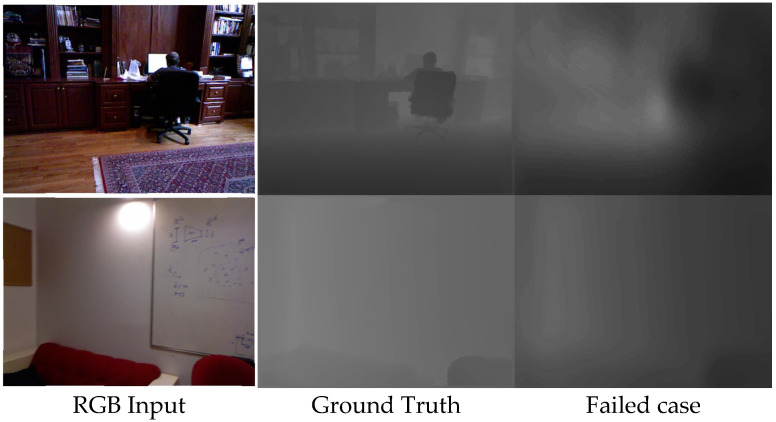
Failed cases generated by the proposed model with the comparison to ground truth.

**Table 1 biomimetics-09-00747-t001:** Detailed architecture of the proposed depth estimation network.

Module	Layer Type	Weight Dimension	Stride
Encoder	Conv	64 × 3 × 9 × 9	1
	ResBlock	64 × 64 × 9 × 9	1
	Conv	128 × 64 × 7 × 7	2
	ResBlock	128 × 128 × 7 × 7	1
	Conv	256 × 128 × 5 × 5	2
	ResBlock	256 × 256 × 5 × 5	1
	Conv	512 × 256 × 3 × 3	2
	ResBlock	512 × 512 × 3 × 3	1
	Conv	512 × 512 × 3 × 3	2
ResNet	6x ResBlock	512 × 512 × 3 × 3	1
Decoder	Upsampling	-	-
	Conv	512 × 512 × 3 × 3	1
	ResBlock	512 × 512 × 3 × 3	1
	Upsampling	-	-
	Conv	256 × 512 × 3 × 3	1
	ResBlock	256 × 256 × 5 × 5	1
	Upsampling	-	-
	Conv	256 × 128 × 5 × 5	1
	ResBlock	128 × 128 × 7 × 7	1
	Upsampling	-	-
	Conv	128 × 64 × 7 × 7	1
	ResBlock	64 × 64 × 9 × 9	1
	Conv	64 × 3 × 9 × 9	1

**Table 2 biomimetics-09-00747-t002:** Performance analysis of state-of-the-art architectures with the proposed network architectures.

Architectures	Eigen et al.	Sihaeng et al.	Zhang et al.	Proposed
RMSE	0.907	0.454	0.590	0.416

## Data Availability

No new data were created or analyzed in this study.

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
