# Peer review of "Deep Neural Networks for Accurate Depth Estimation with Latent Space Features"

_biomimetics, 2024, doi:10.3390/biomimetics9120747_

Round 1

Reviewer 1 Report

Comments and Suggestions for Authors

This paper introduces a method for obtaining depth values from RGB images. In general, the innovation is small, and there are the following points need to be improved:

1.    In the abstract, you built a new loss function to improve the accuracy of depth values,please briefly describe how this loss function works. And you mentioned testing on the NYU dataset, but I think you should add an experiment on the KITTI dataset.

2.    In the introduction section, paragraphs 1,2,3 are too redundant and can be refined, such as reducing the description of techniques such as stereo matching, because your research is a monomural depth estimation, you should focus on what is relevant to your research. In addition, the introduction of monocular depth estimation techniques in paragraph 4 is too old, and some new techniques can be introduced.

3.    In the related work part, the listed research work lacks systematic classification and induction, and is relatively scattered. At the same time, when you introduced each study, you simply described its main content without analyzing the advantages and disadvantages of each study. I suggest that you introduce each study with a brief evaluation of its importance and limitations in the field of depth estimation.

4.    For the methods section, you do not need to describe some of the details of the network, please highlight the description of your own innovation and work. In addition, your schematic looks no different from UNet and does not play a role in explaining your network structure, can you modify it?

5.    In the experimental section, you might consider testing on some other datasets to further verify the generalization ability of the model.  Although good results have been obtained on the NYU Depth V2 dataset, a single dataset may have limitations, and adding more experiments can make the performance evaluation of the model more comprehensive and reliable.

6. Some recent works should be carefully surveyed, inclding "Progressive learning with multi-scale attention network for cross-domain vehicle re-identification","Manifold-based Incomplete Multi-view Clustering via Bi-Consistency Guidance" and "Unsupervised vehicle re-identification with progressive adaptation".

7.    In the conclusion part, you can discuss the limitations of your proposed method and the future research direction, or how to apply your method to related fields, so as to provide some ideas for subsequent research.

Author Response

Review report for “Bioinspired Sensorics, Information Processing and Control”

  • A section of Biomimetics Journal
  • Article titled " Deep Neural Networks for Accurate Depth Estimation with Latent Space Features
  • Authors: Siddiqui Muhammad Yasir1, Hyunsik Ahn2,*

Thank you very much for your work in reviewing our manuscript. Your comments are very valuable and helpful for improving our work. We have studied them carefully and have revised our paper accordingly. Response to your comments and the modifications made are as follows.

Reviewer 2 Report

Comments and Suggestions for Authors

Dear Authors

The manuscript titled  “Deep Neural Networks for Accurate Depth Estimation with Latent Space Features” presents a new depth estimation method using a dual encoder-decoder neural network that improves monocular depth accuracy by refining depth boundaries, achieving high performance on indoor scenes for human-robot interactions.  The paper addresses important issues however it needs improvements.

1.      Improvements required for the abstract

It mentions that the method "sets a new benchmark" but doesn’t clearly specify what this benchmark is, the abstract does not specify which existing methods were compared or how the superiority was measured.

2.      This introduction has some technical issues. The introduction a lacks a clear explanation of the basic ideas and motivation for this study, instead jumping into various details. It introduces many concepts at once without defining how they fit together, such as stereo-matching, structured-light sensors, and statistical feature-based methods, which reduce clarity. The contributions listed at the end are vague and need clearer wording to distinguish how this study advances the field. Adding concise explanations, breaking down the content into simpler sections, and focusing on specific innovations would make the introduction clearer and more accessible.

3.      This literature review needs to increase its clarity and impact. It simply lists previous works without explaining their relevance or how they contribute to the field, making it difficult to understand the connections between them. Each study is introduced with minimal context, and there is no critical analysis of the limitations or gaps in these methods. The review lacks a cohesive structure, presenting the studies in a random order rather than grouping similar approaches or comparing techniques, which would help clarify the trends and challenges in depth estimation. It ends abruptly with a vague transition to the current paper’s contributions without summarizing key takeaways from the reviewed studies. Adding diverse CNN studies can showcase the versatility of CNNs, offering design insights for depth estimation. For example, "3D-CNNHSR: A 3-Dimensional Convolutional Neural Network for Hyperspectral Super-Resolution" shows how 3D-CNNs handle complex spatial data, and “Iebins: Iterative elastic bins for monocular depth estimation” for monocular depth estimation, while "CNN-Based Automated Weed Detection System Using UAV Imagery" demonstrates CNNs’ effectiveness in noisy, real-world conditions. These examples highlight CNN's adaptability, potentially inspiring improvements in-depth estimation techniques.

4.      The architecture relies on a two-network system (color-to-depth and depth-to-depth), this complexity might lead to increased computational demands and longer training times, which can be challenging in real-time applications. The reliance on skip connections in the ResBlocks to capture local details may not be sufficient to address all depth estimation challenges, especially in complex scenes with varying lighting conditions and occlusions. The method also assumes that the latent features learned during training will effectively generalize to unseen environments, which might not always hold true. The loss function, which combines data loss, latent loss, and gradient loss, could become complex, making it hard to balance these different components during training. The use of bilinear interpolation for up-sampling in the decoder may not produce the best quality depth maps, as it can lead to smoothing and loss of important edge information.

5.      An experiment environment with computational complexity, hyperparameter tuning and optimization should be added. See and add CDLSTM; SMOTEDNN.

6.      Rewrite the conclusion for clarity and flow.

Comments on the Quality of English Language

Extensive English editing is required.

Author Response

(The authors gave the same response as above.)

Reviewer 3 Report

Comments and Suggestions for Authors

Thank you for submitting your original paper “Deep Neural Networks for Accurate Depth Estimation with Latent Space Features”. In this time, there are some comments and questions.

(a).  It is not enough to explain the relation among your work and references in “Related Work”.

(b).  Figure 1 is not accuracy. You should revise including block size. Figure 2 is also the same.

(c).  Figure 3 is not accuracy. You should add color bar graph for heat map.

(d).  Figure 4 is not accuracy. You should show the evaluation criteria adding depth bar graph.

(e).  For line 372, there is no caption for Figure. You must revise adding caption appropriately.

Comments on the Quality of English Language

This paper should be checked for the quality of English language by an English native speaker. 

Author Response

(The authors gave the same response as above.)

Reviewer 4 Report

Comments and Suggestions for Authors

Comments (L 154 means line number 154):

L 154-155: "Zhao et. al., [25] Examine "replace by "Zhao et. al. [25], examine ".

L 168: "the robot can understand "replace by "the robot can mimic understanding ".

L 207: enlarge Figure 2 to the paper width.

L 225: Shift Table 1 to keep it compact within one page.

L 236: change “2/2" in the formula (1) according to the formula (4). Probably, you mean the square of the Euclid distance. Let the 2 for square and express the L2 measure by words.

L 240, L250: Subscripts in the text are not formatted correctly. Specifically, in L 240, it may lead to misunderstanding. Follow the formatting used in formulas.

L 292-293: Reorganise Figure 3. into a 2x3 matrix and enlarge the pictures.

L 298-299: there is no explanation for applied hyperparameters. Have any experiments been performed?

L 329-330: Why is the result achieved compared to the worst case?

L 345: The dimensionless RMSE does not allow assessing the solution's suitability for robot-human interactions in general. Is the dimension of the RMSE in meters or millimetres? Also, the training set used is not suitable for all cases of robot-human interactions. Please specify which situations you have in mind.

L 351: Five metrics are indicated, but only one is mentioned.

L372: Add the number and the description of Figure 5 instead of "the figure above" (L 375).

L 374: This is a general remark for misusing the machine-learned models. Generally speaking, anomaly detection must filter the samples that do not match those in the training set.

References:

13, 21, 33 – have no authors,

9, 24, 25 - incomplete references.

Author Response

(The authors gave the same response as above.)

Round 2

Reviewer 2 Report

Comments and Suggestions for Authors

All the comments have been addressed.